# Are the Olfactory Receptors Present at the Sperm Membrane Involved in Reproduction?

**DOI:** 10.3390/ijms241411277

**Published:** 2023-07-10

**Authors:** Francis Galibert, Naoual Azzouzi

**Affiliations:** Institut d Genetique et Developpement de Rennes (IGDR), UMR 6290, Université de Rennes, 35000 Rennes, France; naoual.azzouzi@univ-rennes1.fr

**Keywords:** mouse, olfactory receptors, spermatozoids, reproduction

## Abstract

Olfactory receptors (ORs), key components in ensuring the detection of myriad odorants, are expressed not only on the surface of olfactory neurons but also in many other tissues. In the case of ORs expressed at the sperm membrane, in vitro experiments with human and mouse spermatozoids have shown that they move toward the regions with the highest concentration of bourgeonal and lyral, respectively. However, to date, no in vivo experiment has shown any biological function of these ORs. To demonstrate a possible role in vivo of ORs in sperm chemotaxis, we overloaded the vaginal space of female mice from the prolific Swiss CD1 strain with lyral to induce competition with the supposed natural ligand and to prevent its detection. As shown, the mice that received lyral had much fewer newborns than the control mice treated with PBS, showing that lyral has a strong negative impact on procreation. This indicates that the ORs at the sperm surface are biologically active and make an important contribution to reproduction. Control experiments performed with hexanal, which does not alter sperm movement in vitro, indicate that the inhibition of reproduction observed was specific to lyral. In addition, we show that males are attracted to the smell of lyral, which acts as a pheromone, and prefer to copulate with mice marked on their back with lyral rather than with those that have not been marked. These results suggest an explanation for some cases of human infertility, which could result from an absence of recognition between the natural ligand and the ORs, either due to a mutation or a lack of expression from one of the two partners, allowing for the development of a diagnostic tests. These results might also lead to the development of a novel contraception strategy based on the use of vaginal tablets delivering an odorant or a drug that competes with the natural ligand.

## 1. Introduction

Olfactory receptors (ORs) were discovered by Linda Buck and Richard Axel [1]. This discovery deeply transformed the field of olfaction, and the two authors of this seminal work were honored with the Nobel Prize of Physiology/Medicine in 2004. In their search, Buck and Axel hypothesized that a special category of G-protein-coupled receptors (GPCR) might be expressed by the olfactory neurons present in the nasal epithelium to somehow participate in the recognition of the myriad of chemically different odorant molecules. As the expression of some members of this extended family of GPCRs was initially thought to be restricted to the olfactory epithelium (OE), they were named olfactory receptors (ORs).

However, not long after their identification in rat olfactory epithelium, a number of these genes were shown by polymerase chain reaction (PCR) and cloning to be transcribed in human and dog testes [2]. Critically, Western blotting with antibodies raised against the NH2 terminal portion of the receptor DTMT identified this receptor in protein extracts of spermatozoa prepared from canine ejaculate. This receptor, also named OR1E2, was further localized to the sperm midpiece by immunoreactivity with the same serum, giving weight to the hypothesis that ORs would play a specific role in sperm chemotaxis and fertilization [2,3].

The presence of several receptors was then confirmed and further extended by immunocytochemistry experiments showing that different receptors were detectable on different regions of human spermatozoa [4]. Furthermore, the presence of different OR transcripts was observed in many other tissues, such as a developing rat heart [5], human erythroid blood cells [6], brain [7], prostate [8] and many others. Interestingly, it was noticed that the OR transcripts identified in nonolfactory tissues were not a mere representation of those present in the olfactory epithelium but a subset of the ORs encoded in the genome; even more interestingly, different subsets were identified in different tissues. Later on it was shown that the activation of OR in colorectal cells lead to inhibition of cell proliferation and apoptosis [9], and that OR play in blood pressure regulation [10] or the implication of OR 51B5 in myolenous leukemias cells [11] and more [12,13].

These results suggest an actual biological role for the ORs that was not the result of mere promoter leakage, as first suggested [14]. Along these lines, it was shown that freshly prepared human spermatozoa in a gradient of bourgeonal migrated toward the pole with the higher concentration [15] and that this migration was dependent on adenylate cyclase (mAC), a crucial component of the OR transduction signal machinery [16]. In parallel, it was shown that the addition of lyral a ligand of MOR23, induces an increase in intracellular Ca^2+^ in a fraction of spermatogenic cells prepared from testis and a positive migration of mouse spermatozoa in a gradient of lyral concentration [17]. Furthermore, Ralt et al. [18] showed in an in vitro experiment that preparations of follicular fluid were able to attract freshly prepared human spermatozoids, suggesting that human egg fertilization could be associated with a human follicular factor, acting as ligand of the sperm’s OR(s). Although a possible role of the ORs expressed in the membrane of spermatozoa was evoked very early [2], and although several in vitro observations gave credence to the hypothesis that ORs expressed in tissues other than the OE would have a specific role, no in vivo experimentation confirming or excluding such a role has been published to date, despite the great importance such a demonstration would have.

It is in this context that we chose to perform a simple analysis based on the following hypothesis: if the OR(s) present on the surface of spermatozoa play some role in guiding spermatozoa to reach the egg, we anticipate that overloading the vaginal space with an odorant recognized by these ORs might perturb or even prevent the chemotaxis process and consequently, could alter or prevent reproduction.

The aim of this paper is to present the results of these experiments.

## 2. Results

Three experiments were successively carried out with Swiss CD1 mice, a very prolific strain that have litters of 10 to 15 newborns or sometimes more. lyral and hexanal were used as odorants [17].

To set up the first experiment, mice were divided into two groups: the control group, which received only the solvent, and the second group of mice, which received the odorant. At 5 p.m., 40 μL of solvent alone or 40 μL of diluted odorant were gently injected into the mouse vaginal space. Soon after, each female was individually placed in a cage, and a male was added as indicated in the Methods section. To minimize any adverse effect due to the environment, as well as to note whether the addition of lyral might affect male behavior, an equal number of mice in each group received the odorant (the “test” group) or the PBS alone (the “control” group) on the same day. On the day after injection, the vaginas were inspected to detect any plug that would indicate that copulation took place. Females with no plug and all males were saved for another round of experimentation. Females with a plug were isolated until term delivery, approximately 3 weeks later. At the time of delivery, the number of newborns was recorded. Table 1a summarizes the results.

All female mice that received only PBS had between 10 and 17 newborns except one, which had 6 newborns yielding a total of 385 newborns. Among the female mice that received the odorant, 13 had no pups, 11 had more than 10 pups, and 7 had between 3 and 9 pups, yielding a total of only 177 newborns. Altogether, these numbers indicate that lyral has a strong negative impact on reproduction.

At this stage, we have no explanation concerning the only “control” female mouse having only 6 newborns while all the others had between 10 and 17. Neither the literature nor the records of our animal house mention a similar situation. However, according to Janvier laboratory experts, this might happen from time to time, representing the lower side of a Gaussian distribution. In the absence of an explanation for this small number of pups, we decided to consider all the “test” mice having had pups as unaffected by the injection of lyral, whatever the number of pups.

Thus, we tabulated in the test series 18 females as unaffected by the injection of lyral and the 13 females with no pup as affected. These two numbers when compared with 31, the number of mice in the control series, gave a *p* value of 0.00069 with the Fisher test, indicating that the overloading of the mouse vagina with lyral prevents or limits procreation.

Considering that the seven females with fewer pups (between three and nine) than usually recorded for this mouse strain might have been affected by the presence of lyral, we also performed a Welch’s two sample *t*-test, which considers the whole distribution of newborn numbers (Figure 1a). In this case, a *p* value of 8.2 × 10^−7^ is given, again indicating a strong negative effect of lyral on procreation.

In a second experiment aimed at controlling whether the observed effect is truly due to lyral, this odorant was replaced by hexanal, a component to which mice are sensitive but to which mouse sperm cells do not react [15,17].

The design of this experiment was slightly different in that two female mice were placed in the same cage in the presence of a male. In the vaginal space of one of the two mice, we loaded 40 μL of PBS alone, whereas in the vaginal space of the second, we loaded 40 μL of a hexanal dilution in PBS. As in the previous experiment, we recorded the presence or absence of plugs on the following day. The females with a plug were maintained alone in a cage until delivery, and the females without plug and the males were saved for another round of experimentation.

The results of this second experiment are presented in Table 1b.

In this table, one can observe the following: (1) Out of a total of 20 trios, 9 mice having received lyral had a plug versus 10 mice treated with PBS, whereas in one case only, the two female mice had a vaginal plug. (2) The number of newborns for each pregnant mouse was between 10 and 16, yielding a total of 113 and 131 pups, respectively, indicating no effect of hexanal on procreation as confirmed with the Welch’s two sample *t*-test giving a *p* value of 0.6 (Figure 1b). As such, this result reinforces the specificity of the negative impact of lyral on reproduction as observed in the previous experiment.

We then performed a third experiment that followed the same design as the second experiment, to investigate whether lyral, which attracts sperm cells, also attracts the male, similarly to pheromones. For this third experiment, one female was marked with a drop of lyral on its back, and a second female was marked with a drop of PBS alone. Table 1c summarizes the data.

Out of a total of 31 trios, we observed that 23 of the mice that had pups were marked with lyral on their backs, while the other 8 were marked with PBS. A statistical comparison of these two numbers, gives a *p* value of 2.9 × 10^−4^ (Fisher test), indicating that the males were preferentially attracted by the females marked with a drop of lyral on their back. Again, as previously observed, all pregnant mice had large litters of between 10 and 17 newborns.

## 3. Discussion

The results obtained in these experiments clearly indicate that the presence of lyral in the vaginal space of a mouse is able to impede reproduction. This effect is specific to lyral as neither PBS nor hexanal have any effect. This experiment is, to our knowledge, the first to show in vivo that ORs present at the sperm membrane react to the presence of a ligand, in this case lyral. This result strongly suggests that ORs might play a prominent role in sperm chemotaxis and egg fecundation.

This negative impact on fecundation most likely results from competition between a natural ligand and the odorant. This ligand emitted by the egg or the tissues nearby [12] could be a short fatty acid, as very recently suggested by Teveroni et al. [19]. Mouse behavior and mating in particular are highly dependent on odorants that are sensed by both the ORs present in the main olfactory epithelium and those present in the vomeronasal epithelium. Therefore, one can wonder whether the addition of lyral affected procreation not by impairing sperm chemotaxis but via another route. 

However, since both lyral and hexanal are recognized by the chemosensory systems mentioned above, but only lyral is able to alter chemotaxis and procreation, it is unlikely that the limitation of procreation would be due to a cause other than competition between the lyral and the natural ligand. These results are reminiscent of those observed for progesterone [20], which binds to CatSper [21]. CatSper is a Ca^2+^ channel located on the flagellar midpiece of most mammalian sperm cells. Its activation by progesterone regulates the concentration of intracellular Ca^2+^, which induces sperm accumulation, hyperactivation, capacitation and acrosome reaction [22]. Furthermore, it has been observed that incubating cells with an anti-CatSper1 IgG [23] or immunization of male mice with a DNA vaccine encoding the whole open reading frame of the mouse CatSper1 gene causes a significant reduction in fertility [24]. However, in vitro data concerning chemotaxis toward the egg induced by progesterone are less compelling; some results indicate a chemotaxis effect, while others do not [25]. Interestingly, the third experiment reported here shows that lyral, in addition to modifying sperm mobility, acts as a pheromone by attracting the male when deposited on the back of the female mice.

Considering that human spermatozoa migrate towards the richest pole in bourgeonal [15], just as murine spermatozoa migrate towards the pole enriched in lyral, we hypothesize that bourgeonal or another odorant recognized by human spermatozoa would be able, after vaginal injection, to block the recognition between spermatozoa and their natural ligand and thus prevent procreation, as does lyral injected into the mouse vaginal space. This hypothesis leads to two important applications. The first relates to cases of human infertility which could be due, among many other reasons, to a lack of communication between the natural ligand and the ORs present on the surface of the spermatozoa. This communication defect could itself have several causes, including in particular the existence of deleterious mutations in one of the genes encoding the ligand or one of the ORs. Infertility could also arise due to a lack of expression of either the ligand(s) or the receptor(s). At this point, it is important to recall the cases of human couples encountering sterility problems that could be solved by a direct intracytoplasmic injection of sperm (ICSI), and in which one member of the couple had children with another partner without such challenges. Clearly, these situations suggest a lack of recognition between a ligand and its receptor(s). An exhaustive analysis of the allelic variations in the genes encoding the ORs expressed by spermatozoa among the human population might shed light on some cases of male sterility, leading in the future to an easy diagnostic test. Unfortunately, similar polymorphism analysis of the gene(s) coding for the ligand is presently not possible since the nature of the ligand is unknown.

The second opportunity offered by the present experimental results would be the development of new contraceptive drugs. Bourgeonal, or another product to which human sperm respond, could be given to females via a pharmaceutical preparation such as vaginal tablets or hydrogel. The great advantage of this approach is that it will be based on different principles than the other contraceptive methods, using a completely different pharmacological route with no impact on hormonal metabolism. Such contraceptive drugs would thus avoid the side-effects associated with current contraceptive pills.

However, a new contraceptive drug based on the vaginal delivery of an odorant or an assimilated drug able to perturb the correct movement of spermatozoa toward the egg must be 100% reliable, with no failure. Therefore, the reasons why some female mice treated with lyral became pregnant must be determined. At this stage, we have no explanation, but several hypotheses: for example, the quantity of lyral delivered to some mice may have been insufficient, lyral may not be the optimal chemical, or the timing between the injection of lyral and the copulation may have been inappropriate. These hypotheses must be tested as the next step toward the aim of developing such a drug. But, the fact that a proportion of the mice treated with lyral had a lower than average number of newborns suggests that the amount or conditions of the use of lyral were not optimal to block the entirety of the spermatozoa, leading to a lower level of newborns. 

## 4. Materials and Methods

Lyral (Ref 95594) and hexanal (Ref 115606) were purchased from Sigma–Aldrich. They were diluted in PBS before use to obtain a solution of lyral (0.018 M) and a solution of hexanal (0.03 M).

Five-week-old mice were purchased from Janvier Laboratory and housed in the institute’s animal facility under the supervision of the chief manager throughout the experiments. During this period, females and males were kept apart. After three weeks of adaptation to their new environment, the first round of experiments was started. Some animals were utilized up to the age of 48 weeks. Female mice received at 5 p.m., via the intravaginal route, 40 μL of the diluted odorant solution or of PBS solution with the aid of a nonsurgical embryo transfer device from ParaTechs, Lexington, KY, USA (60010 mNSE) and were placed in a cage alone. A few minutes later, a male was added to the cage. The next morning, male mice were isolated and saved for another experiment a week or so later. Vaginas were inspected to detect any plug that would indicate that copulation took place. Females with no plug were reserved for another experimentation one or two weeks later. Females with a plug were isolated until term delivery, 3 weeks later. At the time of delivery, the number of newborns was recorded.

Animal welfare as well as any potential negative effects due to lyral or hexanal injection were regularly monitored throughout the experimental process. In the case that a problem was observed, the experiments would have been interrupted to limit any suffering of the animals, but this situation never occurred. Moreover, we noted no difference in the behavior of the females that participated in several rounds of experimentation and were injected either with the lyral or hexanal solutions.

## Figures and Tables

**Figure 1 ijms-24-11277-f001:**
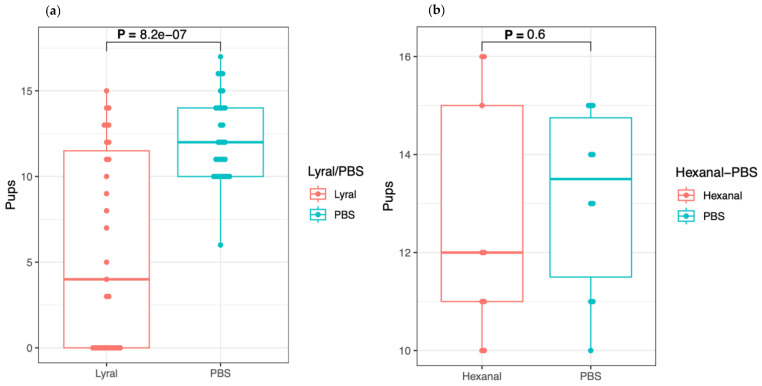
Distribution of the number of newborns obtained with the Lyral and Hexanal experiments. These figures show the distribution of the number of newborns (Table 1a) of the mice treated intravaginally with Lyral (“test” series) versus those in the “control” series that received PBS. (**b**) corresponds to the second experiment (Table 1b) during which the mice received Hexanal instead of Lyral. In the (**a**,**b**) the Y axis correspond to the number of newborns, and the horizontal bars to the medians. Statistical treatment of the data was made with the Welch two sample *t*-test giving the respective values of *p* = 8.178 × 10^−7^, *p* = 0.5966.

**Table 1 ijms-24-11277-t001:** Litter size. Each of the 3 panels reports the number of newborns obtained in one of the 3 experiments. Panel (**a**) gathered the data obtained by intravaginal injection of 40 µL of Lyral. Panel (**b**) groups those obtained with Hexanal and panel (**c**) reports the results obtained with Lyral when the latter was deposited on the back of the “test” mice and not in the vagina; For the 3 panels, columns 1 and 3 correspond to the identity of the “test” and “control” mice respectively, columns 2 and 4 give the size of the litters. For the three tables, the order of the data corresponds to that in which the mice were treated; The statistical analysis of the data (panel (**a**,**b**)) made with the Welsh two sample *t*-test gives the respective values of *p* = 8.2 × 10^−7^ and *p* = 0.6 (Figure 1). These *p* values clearly indicate the negative and specific impact of Lyral on procreation. Comparison of the number of pregnant mice 23 marked on their back with Lyral to the number of pregnant mice 8 marked with PBS (**c**) gives a *p* value of 2.9 × 10^−4^ with the Fisher test.

(**a**)				(**b**)				(**c**)			
**LYRAL**		**PBS**		**Hexanal**		**PBS**		**Lyral**		**PBS**	
**Mouse ID**	**Pups**	**Mouse ID**	**Pups**	**Mouse ID**	**Pups**	**Mouse ID**	**Pups**	**Mouse ID**	**Pups**	**Mouse ID**	**Pups**
602	12	585	10	746	11			740	15		
589	11	588	10	732	12			739	11		
591	0	597	12			747	15	711	10		
598	10	621	12	788	11			737	15		
592	0	586	14	827	10					741	15
590	4	593	16			777	11			728	10
611	11	622	11			729	15			730	11
597	0	587	15			812	13	754	10		
616	0	668	11	832	16			729	12		
623	3	652	14			813	13	734	16		
613	0	677	14			790	10	753	11		
614	14	680	13	824	12			733	14		
615	15	650	6	813	10			752	11		
618	0	699	14			786	11	743	16		
591	0	695	14	815	15			730	15		
688	8	696	17			820	14	740	15		
687	0	676	15			851	15			738	14
682	14	678	10	356	16			756	13		
684	0	697	12			854	14			778	15
683	9	712	10	Total	113		131			774	14
685	0	713	11	Mean	12.5		13.2	763	15		
681	3	704	10	Median	12		13	751	12		
645	13	705	12					725	14		
646	0	706	16					750	16		
647	13	708	16					759	17		
648	13	709	11					773	11		
653	0	710	13					757	13		
684	0	711	10							776	16
716	5	714	11					775	15		
717	12	721	14					771	12		
718	7	738	11							744	14
Total	177		385					Total	299		109
Mean	5.7		12.4					Mean	13		13.6
Median	3		12					Median	13		14

## Data Availability

All data will be freely available upon publication. Any further requests can be obtained from the corresponding author galibert@univ-rennes1.fr.

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
