# Peer review of "Are the Olfactory Receptors Present at the Sperm Membrane Involved in Reproduction?"

_ijms, 2023, doi:10.3390/ijms241411277_

Round 1

Reviewer 1 Report

This manuscript describes the possible role of olfactory receptors in the fertility of mice. The experiments are well done but the presentation of results could be improved by using only the figure for the first experiment but the tables could be used for the second and third experiment to emphasize that only 1 of 2 females got pregnant.  Rather than the extensive description of results in the tables just state the overall means and standard errors of confidence rages.  Also just use p<0.001 etc and do not give exact p levels as these can be confusing.

The conclusion following the discussion is missing.  Also, the conclusion in the abstract should not include the speculation on the human as you have no data on this.

Author Response

Reviewer 1 asks to delete table 1 which gives the number of newborns that each female mice, “test” or “control’, had considering that Figure 1 built with these data would be sufficient. First of all, we completely disagree with this opinion insofar as the number of newborns per female treated with lyral or PBS corresponds to the essential part of the work and the scientific message that results from it. In addition, we note that the other reviewer, far from finding this superfluous table, asks that other data should be added to it.

As requested, we have reduced in the legends the description of the results.

Reviewer 2 Report

The manuscript presents the data about in vivo verification of the hypothesis that olfactory receptors are involved in reproduction. The study is very interesting and concerns an attractive, novel subject of the role of olfactory receptors present on sperm cells membrane. It provides a good complement of previous in vitro studies and the results have a potential for practical implementation (contraception, infertility diagnosis), although, as the Authors rightfully discussed, more research in this field is required.

The experiments were quite simple (no advanced methods applied), but sufficient to reach the study goal. Paper type ‘communication’ is suitable.

The experiments were well designed and the results are presented clearly, however, the style and the language of the manuscript could be improved for better scientific soundness. E.g. a scientific name for mice ‘babies’ is ‘pups’ and instead of ‘many fewer’, ‘significantly fewer’ should be used. ‘Spermatozoids’ are gametes of lower plants, in mammals proper name is “spermatozoa” etc.

The structure is incorrect – a huge part of the results (Lines 91-105; 164-173; 200-204)  should be in ‘material and method’ section.

I have also some doubts about the statistics. Shapiro-Wilk test is used to test data normality, not to compare groups. If data in the test group were not distributed normally, differences between groups should be analyzed by U Mann-Whitney test.

Additionally I have some minor remarks to the manuscript. After revision, I find the manuscript worth publishing.

Minor remarks:

1.     Title: at the present form literally it suggest that any receptor can be involved olfactory. I propose little change: Are the olfactory receptors present at the sperm membrane involved in reproduction?

2.     Abstract: results of the third experiment should be included, as well as some numerical results.

3.     Line 61: dot after [8] to be removed.

4.     Line 43: first full name of GPCR, then an abbreviation.

5.     Line 92: only two experiments were with Lyral, this part should be removed or hexanal should be mentioned.

6.     Table 1: Total and average number of newborns should be added to table 1, similarly to two other tables. If data were not distributed normally, median should be used instead of mean value in all tables.

7.     Table 1: ‘we never observed for the mice having less than 10 babies any dead fetus the presence of which would indicated some suffering during pregnancy’ – how Authors checked all mice for the presence of dead fetuses? You mean abortion or stillbirth? What about possibility of eating dead newborns? Or embryo resorption in early pregnancy? There is too many possible explanations for this case, I would remove this statement.

8.     Figure 1: a mouse in control group with 6 offspring is not marked 

9.     Table 2: Why the ID numbers and the litter sizes for the trio in which the two females had a plug are not reported in this table? If the authors decided to remove this trio, then better to state total number of 19 trios, otherwise this phrase: ‘Of a total of 20 mouse trios, 9 females receiving hexanal were pregnant, compared to 10 females that received PBS.’ is not mathematically correct.

10.  Tables: There is no explanation for the order of mice in tables – if no special reasons, they can be listed in order from smallest ID number to largest

11.  Tables: I don’t think such extensive explanation in table’s footnotes is necessary.

12.  Line 270-273: ICSI solves many infertility problems, not only lack of expression of receptor/ligand. I would be really caution and limit this only to unexplained cases, where no other semen abnormalities were detected and no reasons for infertility were diagnosed.

13.  Line 310 – a total number of mice used in experiment should be included.

14.  Line 322 – how 5 weeks old mice become 48 weeks old? Did the experiment last almost one year?

15.  Line 449: a typo, ‘Teves’, not ‘eves’

Author Response

1) In his report, reviewer 2 requests that lines (91-105; 164-173; 200-204) be moved to the materials and methods section. In fact, these lines correspond to the legends of figures or tables and as such they have no place in the body of the text. In reality their presence in the body of the text is due to an error which occurred during the loading of the manuscript. This has been corrected.

2) The use of the Shapiro-Wilk test was criticized. After questioning a statistician of the Institute, we replaced the Shapiro-Wilk test by the Welch Two Sample t-test

 Minor comments

 All the requests were taken into consideration and corrected, as the comparison of the old and new document can show. With regard of the quality of English we have corrected some terms which were pointed out by the reviewer and a number of phrasing were rewritten. Although we are aware that the English used is surely not the most elegant we believe that there should be no grammatical errors because it was edited by Springer Nature editing service before loading.
